# Redox Homeostasis and Non-Invasive Assessment of Significant Liver Fibrosis by Shear Wave Elastography

**DOI:** 10.3390/diagnostics14171945

**Published:** 2024-09-03

**Authors:** Anna Egresi, Anna Blázovics, Gabriella Lengyel, Adrienn Gréta Tóth, Barbara Csongrády, Zsuzsanna Jakab, Krisztina Hagymási

**Affiliations:** 1Department of Surgery, Transplantation and Gastroenterology, Semmelweis University, 1091 Budapest, Hungary; egresi.anna@semmelweis.hu (A.E.); lengyel.gabriella@semmelweis.hu (G.L.); 2Department of Surgical Research and Techniques, The Heart and Vascular Center, Semmelweis University, 1091 Budapest, Hungary; blazovics.anna@semmelweis.hu; 3Centre for Bioinformatics, University of Veterinary Medicine, 1078 Budapest, Hungary; tothadrienngreta@gmail.com; 4Department of Radiology, Semmelweis University, 1091 Budapest, Hungary; csongrady.barbara@semmelweis.hu; 5Department of Internal Medicine and Oncology, Semmelweis University, 1091 Budapest, Hungary; jakab.zsuzsanna@semmelweis.hu

**Keywords:** chronic liver diseases, liver cirrhosis, liver stiffness, cytokines, leptin, adiponectin, oxidative stress

## Abstract

Hepatic fibrosis with various origins can be estimated non-invasively by using certain biomarkers and imaging-based measurements. The aim of our study was to examine redox homeostasis biomarkers and liver stiffness measurements for the assessment of significant liver fibrosis in different etiologies of chronic liver diseases. A cohort study consisting of 88 chronic liver disease patients of both sexes (age 49.1 ± 14.7 years) was performed. Cytokine profiles as well as redox homeostasis characteristics were determined. Liver fibrosis stages were assessed with shear wave elastography. The plasma levels of four cytokines showed no significant alteration between the four fibrotic stages; however, higher values were measured in the F2–4 stages. Free sulfhydryl group concentration, the marker of redox homeostasis, was lower in significant fibrosis (F0–F1: 0.36 ± 0.06 vs. F2–4: 0.29 ± 0.08 mmol/L, *p* < 0.05). Higher chemiluminescence values, as free radical–antioxidant parameters, were detected in advanced fibrosis stages in erythrocytes (F0–F1: 36.00 ± 37.13 vs. F2–4: 51.47 ± 44.34 RLU%). These data suggest that oxidative stress markers can predict significant fibrosis, with the aim of reducing the number of protocol liver biopsies in patients unlikely to have significant disease; however, their role in distinguishing between the certain fibrosis groups needs further studies.

## 1. Introduction

Chronic liver disorders are a significant public health issue. Due to various etiological factors (viral infections, metabolic abnormalities, unhealthy nutrition, drug-induced toxicity, and autoimmunity), liver injury appears, which develops as fat accumulation in hepatocytes, necroinflammation, and connective tissue remodeling in liver tissue, resulting in cirrhosis, hepatic failure, hepatocellular carcinoma formation, and, finally, death. The process of progression is complex, multi-pathed, and multifactorial. The evaluation of liver fibrosis is essential for the assessment of liver injury and to predict the prognosis. For these purposes, liver biopsy has been the accepted gold standard for decades. During the histological examination, the METAVIR scoring system is used to assess the extent of inflammation and fibrosis. The stage represents the amount of fibrosis (F0: no fibrosis; F1: portal fibrosis without septa; F2: portal fibrosis with a few septa; F3: numerous septa without cirrhosis; F4: cirrhosis) [1].

Recently, non-invasive approaches employing serological biomarkers, scores, and ultrasound-based techniques have been established for the assessment of liver fibrosis and to follow its progression. Non-invasive methods also have a role in predicting complications and clinical outcomes (esophageal varices formation or hepatocellular carcinoma development) and help to make therapeutic and diagnostic decisions [2,3].

By using certain biomarkers, the degree of connective tissue transformation can be estimated. These direct marker molecules are matrix constructs, which mainly originate from activated Kupffer cells (N-terminal pro-peptide of collagen type III, glycoprotein YKL-40), or they accumulate due to hepatic stellate cell dysfunction and may also be enzymes and cytokines (TGF-ß, IF-γ, and IL-6 and 8) [4,5,6,7,8].

Additionally, the so-called indirect markers, or laboratory parameters used in everyday practice, as members of scores and indices (AAR, APRI, GUCI, FiB-4, etc.), can be helpful as well. Approximately 20–30 multifactorial scores or indices are known in the literature [9].

The main limitation of all these markers is that they are not specific for the liver, and they can be released in other tissue injuries. In the detection of moderate fibrosis (METAVIR stage 2), they are not accurate, and their level is affected by renal function, postprandial status, and extrahepatic fibrogenesis [10].

Chronic low-grade inflammation and TGF-β1 signaling play significant roles in fibrosis formation. The current evidence demonstrates as well that oxidative stress and the antioxidant system may also be pivotal for fibrosis evolution and progression. The liver is one of the key organs where free radicals act. Biomarkers of oxidative stress can also be taken into account, such as lipid peroxidation endproduct malondialdehyde (MDA) concentration and antioxidant enzyme superoxide dismutase (SOD) activity, which were found to correlate with fibrosis [11,12]. However, the levels of redox homeostasis parameters in liver cirrhosis patients are conflicting [13].

Additionally, new and modern imaging procedures are available to characterize liver fibrosis. An FDA-approved technique that uses ultrasonography imaging to measure liver stiffness is shear wave elastography. The shear wave elasticity imaging (SWEI) technique measures the spread of shear waves in the liver tissue. A special software implanted into a conventional ultrasound machine is applied to characterize liver stiffness. The degree of connective tissue formation in the liver can be classified into four stages (from F0–F1 to F4, ≥F2 with significant fibrosis) based on the liver stiffness [14]. Elastography-based liver fibrosis markers are extensively used in clinical practice due to their reliability and validity [2,15].

To reduce the need for invasive liver biopsies and raise diagnostic accuracy of these non-invasive methods, score systems (Fibrotest, Hepascore, ELF test, etc.) [10,16], sequential algorithms (SAFE biopsy, Fibropaca, Leroy) [17], and the combination of scores and imaging techniques (Bonacini formula, PLF score) were created [18,19]. With the help of these algorithms, scoring systems, or combined methods, we could avoid 30–80% of liver biopsies with higher diagnostic accuracy than non-invasive methods alone [20].

The aim of our study was to find a non-invasive redox homeostasis biomarker for the detection of significant liver fibrosis, determined by shear wave elastography.

## 2. Materials and Methods

Hydrogen peroxide, 1,1-diphenyl-2-picrylhydrazyl, luminol, microperoxidase, TNF-α, IL-6, leptin, and adiponectin ELISA kits were obtained from Sigma-Aldrich (St. Luis, MI, USA). Routine laboratory kits were purchased from Beckman Coulter (Brea, CA, USA). The CHR hemoglobin reagent solution was purchased from Reagents Ltd., Budapest, Hungary. Standard solutions, nitric acid, hydrogen chloride, and all the other reagents in analytical grade were purchased from Reanal (Budapest, Hungary).

A total of 88 patients were included (male = 36, female = 52; mean age = 49.1 ± 14.7; BMI = 29.26 ± 5.71 kg/m^2^; ALT = 53.61 ± 34.12 U/L; AST = 57.48 ± 37.25 U/L; GGT = 145.94 ± 164.49 U/L; albumin concentration = 43.73 ± 5.71 g/L; total bilirubin concentration = 17.9 ± 13.88 µmol/L; INR = 1.19 ± 1.02; platelet count = 210.15 ± 87.78 G/L) with chronic liver disease of various etiologies (HCV = 10, HBV = 6, HCV/HBV = 1, NAFLD/NASH = 41, ALD = 19, AIH = 11), selected from the hepatological outpatient department of Semmelweis University. The NAFLD/NASH terms, instead of the nowadays used metabolic-associated steatotic liver disease, were used in this article because, at the time of examination, they were the medical terms. A total of 28/88 (31.8%) patients had hypertension, and 11/88 (12.5%) patients had diabetes mellitus. In the ALD group, the daily alcohol consumption was more than ≥20 g/day for female patients and ≥30 g/day for male patients. Patients with a history of acute cholecystitis, cholangitis, liver neoplasm, or previous malignancy history and/or therapy were excluded.

Liver stiffness (LS) was determined by median values of elasticity by a Toshiba ultrasound instrument (Toshiba Aplio 500, Toshiba Medical Systems, Otawara, Japan), with the combination of elastography with grayscale ultrasound imaging. Examinations were performed on patients on an empty stomach in a supine lying position with the use of a convex ultrasound head in the right subcostal view in a breath-holding condition to avoid motion artifacts. The sampling box was placed in the right lobe of the liver, below the capsule with 1 cm, avoiding the main vessels. For each patient, ten measurements were performed according to the manufacturer’s instructions; values were expressed in kPa (kilopascal), and the median value was calculated. LS measurements with an interquartile range (IQR) lower than 30% (IQR is the difference between the 75th and 25th percentile) were considered to determine significant (≥F2) or advanced (F3–4) fibrosis [21]. The cutoff values of the fibrosis groups were as follows: F0–F1: <7 kPa; F2 = 7–9 kPa; F3 = 9–12 kPa; F4 > 12 kPa, provided by the manufacturer’s instructions based on Ferraioli et al. [22]. The inter-observer agreement with the SWE technique was 95% (CI = 0.91–0.97), and the intra-observer variability was 2% (CI = 0.97–0.99).

Blood samples were collected in Vacuette^®^ tubes (Greiner Bio-One, Monroe, NC, USA) with sodium citrate solutions in concentrations of 3.2%. Plasma was separated from the erythrocyte fraction by the standard method and stored at −20 °C for the measurements.

The erythrocytes were isolated from the plasma and buffy coat by centrifugation at 2500 rpm for 10 min. The erythrocytes were washed with isotonic saline solution (0.9% NaCl, Fresenius Kabi, Graz, Austria) three times, and the hemoglobin content was determined with a CHR hemoglobin reagent. The absorbance of the samples was measured at 540 nm on a Hitachi U-2000 spectrophotometer, and the samples were standardized to 1% hemoglobin [23].

After the blood separation and hemoglobin standardization process, the samples were stored at −20 °C. The cytokine and the redox parameter measurements were carried out within six months.

The plasma levels of four cytokines, leptin, adiponectin, IL-6, and TNF-α, were measured with enzyme-linked immunosorbent assay (ELISA) kits (Sigma-Aldrich^®^, St. Luis, MI, USA) based on the manufacturer’s instructions. The absorbance levels were read at a wavelength of 450 nm with a Microplate Reader (Statfax 2000, Awareness Technology, Inc., Palm City, FL, USA), and the samples were evaluated.

The free sulfhydryl group concentration was measured by Ellman and Lysko’s method with 5,5′-dithiobis-nitrobenzoic reagent in Na-phosphate buffer (pH 7.4) at 512 nm. As a standard, we used reduced glutathione [24].

The H-donor activities of the samples were determined according to the method of Hatano, with 1,1-diphenyl-2-picrylhydrazyl (DPPH) used as a stable free radical by the spectrophotometer. The absorbance of the methanolic DPPH dye was measured at 517 nm. The activity was determined as inhibition %. Inhibition % = [Abs (control) − (Abs (sample) – Abs (sample without DPPH)]/Abs (control) × 100 [25].

The total scavenger capacity (or chemiluminescent intensity) was measured by Berthold’s Lumat 9501 luminometer on the basis of Blázovics and Sárdi’s method in the H_2_O_2_/OH–microperoxidase-luminol system and was expressed in relative light units (RLU%) of standard light [26].

The statistical analysis was performed using the R 4.1.3 program version (Statistica 12.0.0.). A Wilcoxon rank sum test was carried out. The Benjamini–Hochberg method was used to calculate the adjusted *p* value. Significant correlations were determined at *p* < 0.05. Data are expressed as mean ± SD.

## 3. Results

Patients of both sexes were grouped by fibrosis stages characterized by the liver stiffness, measured by shear wave elastography. A total of 50.79% of the patients were classified as stage F0–F1 (no or non-significant fibrosis). The prevalence of significant fibrosis was 49.2% (≥F2), and 19.05% of patients were classified as stage F2, 14.29% as F3, and 15.87% as F4. The results were focusing on F0–1 vs. F2–4, which is of most clinical significance (no and non-significant fibrosis vs. significant fibrosis that demands intervention).

There was no significant difference among the fibrosis groups regarding the measured concentration of the four cytokines (leptin, adiponectin, IL-6, and TNF-α) in the plasma samples, but in the F2–F4 groups, there tended to be higher values of all the measured cytokines than in the F0–F1 (Table 1).

Plasma free sulfhydryl group content, characterizing the main non-enzymatic antioxidant defense, was significantly lower in the significant fibrosis F2–F4 groups compared to the F0–F1 patients (*p* = 0.005862, *p*-adjusted = 0.041034). It tended to be lower in the significant fibrosis groups, without statistical significance, as shown in Figure 1.

The hydrogen-donating ability of plasma as a compound non-enzymatic antioxidant marker was not significant between the F0–F1 and F2–F4 stages. The F3 and F4 groups tended to be smaller in value, as shown in Figure 2. The same measurement in erythrocytes showed no difference among the groups (F0–F1 = 21.68 ± 5.98; F2 = 24.21 ± 9.17; F3 = 22.50 ± 10.81; F4 = 20.76 ± 5.19; F2–4 = 22.57 ± 8.10 inhibition%).

We found no difference with induced chemiluminescence (RLU%) as a free radical–antioxidant balance parameter in plasma and erthyrocytes among the patients. In tendency, increased induced chemiluminescence and relative light unit intensity, indicating decreased total scavenger capacity, could be detected in the significant fibrosis stages in erythrocytes, demonstrating the longer-term pro-antioxidant balance in comparison with plasma total scavenger capacity, however without significance in our pilot experiment, as shown in Figure 3 and Figure 4.

## 4. Discussions

Chronic liver diseases are a considerable health problem worldwide. Liver biopsy is the gold standard method for the determination of the hepatic injury, the disease progression, or the stage of liver fibrosis. Because of its invasiveness, it has many limitations, is painful, has many contraindications, and the risk of complications, sampling errors, and inter-observer variability is relatively high [1]. An increasing number of non-invasive markers have been investigated to predict the severity of liver fibrosis in the past decades [27,28,29,30]. Early diagnosis of liver fibrosis is principal to stopping the progression to cirrhosis and hepatocellular carcinoma formation [17]. Various biomarkers characterizing liver fibrosis non-invasively produced conflicting results with contradictory outcomes [17,31]. Many researchers focus on the investigation of non-invasive biomarkers, and new results are published all the time [2,15].

In patients with liver cirrhosis, increased prooxidant markers (serum MDA) and decreased antioxidant capacity (red blood cell catalase, SOD, reduced blood GSH) were measured. The membranes of the red blood cells also vary depending on the redox status in cirrhotic patients, which is related to the higher levels of nitric oxide. The abovementioned differences have been correlated with the Child–Pugh score [32,33]. Significant abnormalities in redox homeostasis of cirrhotic patients were suspected in the background of endothelial dysfunction, responsible for comorbidities and an unfavorable prognosis for such patients as well [13].

Tissue stiffness is a promising biomarker for the assessment of fibrotic tissue accumulation, and ultrasound elastography is a favorable, non-invasive technique to determine a liver fibrosis stage. The 2D-SWE has been shown to correlate well with fibrosis grade in chronic liver disease patients [34], with sensitivity and specificity of 0.85–0.79 for patients with ≥F2, 0.87–0.84 for ≥F3 patients, and 0.88–0.91 for F4 patients [35]. The Society of Radiologists in Ultrasound recommended a “Rule of Four”. If the stiffness values (1) ≤5 kPa: high probability of being normal; (2) <9 kPa: exclude compensated advanced liver disease (cACLD) in the absence of other known clinical signs; (3) 9–13 kPa values are greatly significant of cACLD, but there is need for further verification; (4) >13 kPa are highly suggestive of cACLD; and (5) >17 kPa indicate the presence of clinically significant portal hypertension, with the presence of esophageal varices and the indication of gastroscopy [36]. Screening a large population by liver biopsies or monitoring disease changes with this invasive procedure is unsuitable. It is highly discussed if shear wave elastography can be a gold standard for the diagnosis of the fibrosis stage. There are some limitations. By using SWE techniques, adherence to a strict protocol is important in the decrease in inter- and intra-observer variability. Furthermore, inflammation and, to a lesser extent, steatosis can cause a confounding effect on liver stiffness examinations. The performance of elastography is better for diagnosis in advanced liver fibrosis stages (F3–4) as compared to early stages (F1–2). Elastography is less accurate to distinguish the early fibrosis stages (F1 vs. F2) [34]. Reduced accuracy is observed in lower fibrosis stages (F0–F2) (an accuracy of 72.6 for F0–F1, 63.1 for F2, 83.1 for F3, and 90.5 for F4) [37].

In addition, cutoffs for staging liver fibrosis are system-specific [28,38,39]. In summary, liver biopsy appears to remain the gold standard for staging liver fibrosis, but for disease monitoring, SWE is recommended as a non-invasive tool in patients with chronic liver disease [28].

We investigated four “driver” cytokines and three redox oxidative stress biomarkers in the distinction of fibrosis stages, differentiated by two-dimensional shear wave elastography results, to find a non-invasive, reproducible biochemical, pathomechanism-based fibrosis marker for liver diseases with various etiologies.

The association between leptin levels and nonalcoholic fatty liver disease (NAFLD)-related liver fibrosis is not well proven. Leptin regulates lipid accumulation, plays a central role in inflammatory and immune processes, and modulates angiogenesis by paracrine or endocrine mechanisms [40]. Leptin is significant regarding liver fibrogenesis by the over-expression of profibrogeneic cytokines and transforming growth factor-β. Leptin has immune modulatory and prooxidative properties on lymphocytes. Leptin augmented healthy and HCV lymphocyte proliferations, increased their free radical production, and decreased their antioxidant defense by a diminished level of reduced glutathione [41]. In our pilot study, non-significantly elevated levels of leptin in significant hepatic fibrosis (>=F2) were detected. However, circulating leptin levels were not changed in NAFLD patients with and without fibrosis in a 2024 meta-analysis [42].

Previously, higher levels of adiponectin with antifibrotic properties were demonstrated to be related to the evolution of liver fibrosis, suggesting the role of adiponectin as a non-invasive biomarker for the amelioration of liver fibrosis [43]. The role of adiponectin in carbohydrates and lipid metabolism is well proven. Moreover, its hepatoprotective and anti-inflammatory properties were also demonstrated [44]. Adiponectin suppresses the proliferation and migration of hepatic stellate cells. Furthermore, adiponectin diminishes the expression of fibrogenic genes, such as the connective tissue growth factor, provoked by transforming growth factor-β1 [45,46]. Adiponectin reduces iNOS/NO system activation through adipoR2-AMPK-JNK/Erk1/2-NF-κB signaling, resulting in NO release and further suppressing hepatic stellate cell function and fibrogenesis [47]. The adiponectin level showed no significant correlation with fibrosis stages in our study, with higher levels in the F2–4 stage vs. F0–F1. A recent meta-analysis did not demonstrate a difference between NAFLD patients with or without fibrosis regarding circulating adiponectin levels [48].

Cytokines are inflammatory mediators, playing an important role in the pathomechanism and progression of many acute and chronic diseases, including liver disorders. Earlier studies showed a good correlation between cytokine levels and fibrosis stages; higher levels of IL-6 and TNF-α were associated with the presence of advanced liver fibrosis. However, several studies have not detected associations between cytokine levels and the severity of the disease, as well as the presence of co-morbidities [49,50]. We also found elevated levels of IL-6 and TNF-α in the fibrosis groups (F2–F4), but these results were not significant. The results of circulating levels of inflammatory cytokines as predictors of disease are conflicting; therefore, they will not be sensitive or specific enough to be a reliable liver fibrosis biomarker.

Hepatic stellate cell activation is the key event of liver fibrosis development. Their activation provokes an immune response by cytokine and chemokine release through interactions with immune cells. Their activation is induced by reactive oxygen species (ROS) released from Kupffer cells, hepatocytes, and cytokines. ROS are produced mainly via the mitochondrial electron transport chain or via activation of cytochrome P450, nicotinamide adenine dinucleotide phosphate (NADPH) oxidase, xanthine oxidase, or mitochondrial damage [51].

Glutathione is the key non-enzymatic regulator of the intracellular redox state. It is omnipresent in all cell types. GSH is primarily produced in the liver, and it has a constantly high level in most tissues. GSH is oxidized to GSSG during its antioxidative action; therefore, the GSH/GSSG ratio is a reliable marker of tissue and blood prooxidant status. There are many simple, sensitive, and low-priced, frequently used techniques to measure glutathione concentration in biological samples [52]. In our study, plasma free sulfhydryl group content, determined by spectrophotometry, characterizing non-enzymatic antioxidant defense, was significantly lower in significant fibrosis F2–F4 groups compared to F0–F1. Reduced glutation exerts its antioxidative effect by neutralizing free radicals and peroxides, and there is growing evidence that GSH plays a significant role in cell signaling, affecting many important cell functions like growth, proliferation, and apoptosis. Reduced glutation is considered a natural storage of intracellular NO by the formation of S-nitrosoglutathione. GSH structurally related compounds, as the precursors of glutathione synthesis, N-acetyl-L-cysteine (NAC), S-Nitroso-N-acetylcysteine (SNAC), S-adenosyl-L-methionine (SAM), and S-allylcysteine (SAC), are suggested to be hepatoprotective and have been used in clinical settings for the treatment of hepatic fibrosis [53].

Decreased total scavenger capacity, as a marker of prooxidant–antioxidant balance, could be detected in significant fibrosis stages in erythrocytes, however without significance in our study. The exact role of free radicals in human inflammation in both acute and chronic conditions has never been clearly defined, due to the complexity of the determination of free radical reactions and oxidative damage in humans. One of the two main explanations for that, on the one hand, is the quick reactivity of free radicals; therefore, free radicals are not demonstrable by direct measurement, and their harmful consequences, such as lipid, protein, and DNA peroxidation, are determined by indirect methods. On the other hand, the redox state at the cellular, tissue, and organism levels is highly difficult and hard to assess by a single method. There is no standardized measurement of free radicals and antioxidative balance in humans, and the so-called biomarkers of oxidative stress do not assess oxidative stress accurately, which is why they cannot be used routinely in human disorders. Prooxidant and antioxidant marker levels could also be different depending on the site of sampling (liver tissue, blood, plasma) in a liver patient. The levels of redox parameters could be influenced by many drugs, dietary factors, and chronic diseases that are commonly seen in most of the chronic liver diseases as well [52].

## 5. Conclusions

The development of non-invasive biomarkers of disease has become a major focus of interest in liver diseases to assess the stage, the progression, or to predict the prognosis and the risk of complications. The large prevalence of liver disorders and the invasiveness of a liver biopsy mean that it is impractical. Serum markers of inflammation, apoptosis, oxidative stress, and fibrosis have been broadly investigated. Imaging and ultrasound-based techniques, such as elastography, are becoming more confirmed as non-invasive methods of detecting fibrosis in a variety of chronic liver diseases.

In our pilot experiment, decreased levels of redox homeostasis markers (free sulfhydryl group concentration, total scavenger capacity) could be detected in significant liver fibrosis stages determined by shear wave elastography. Our findings support the importance of these pathological processes in the progression of liver diseases; however, the results should be cautiously interpreted due to the modest sample size and the lack of liver biopsies and histological examinations. In our actual clinical practice, a liver biopsy is performed only when it is necessary for the diagnosis of liver disease. The next main limitation of our study is the heterogeneity of the liver fibrosis etiology, which might result from viral infection, alcohol consumption, autoimmune processes, and metabolic reasons, affecting the inflammatory, free radical status, and liver fibrosis evaluation.

Our findings underline the possibility of the theory of a redox fibrosis where the cellular oxidant and antioxidant systems could be potential therapeutic targets [54]. As more treatment options develop, a non-invasive test is desirable for serial quantification of liver fibrosis; however, liver biopsy still has a role in cases of inconclusive findings or when the non-invasive test results do not correlate with clinical findings. Novel evidence predicts that combinations of these measurements may be promising to avoid invasive histological sampling with complications with serious morbidity and mortality and as a cost-effective marker to identify early fibrotic stages [55].

Presumably, free radical reactions and prooxidant–antioxidant balance in the liver cannot be trustworthily characterized by only one marker. Combinations of redox markers with cytokines, chemokines, and adipocytokines and mergers of serum biomarkers with imaging techniques, like elastography, with the use of equations and algorithms are needed to follow the progression of liver diseases non-invasively to decrease the number of protocol biopsies in patients unconvincing to have significant disease. Newer methods such as proteomics and glycomics allow for the identification of novel markers and may also potentially contribute to our understanding of the pathogenesis of the condition.

## Figures and Tables

**Figure 1 diagnostics-14-01945-f001:**
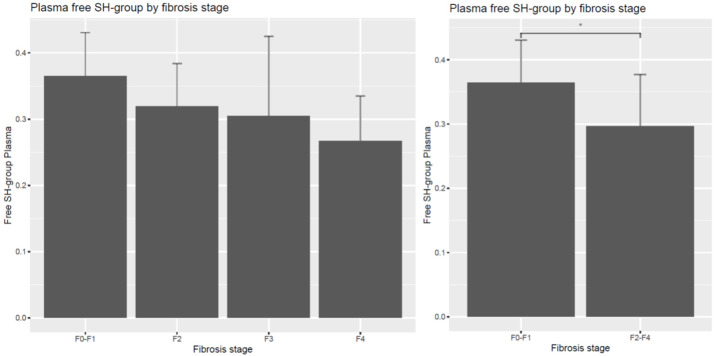
Free sulfhydryl group content (mmol/L) in plasma according to liver fibrosis stages. Decreased free SH group concentrations were measured in the significant fibrosis (F2–4) groups in comparison with the F0–F1 patients without fibrosis. * *p* < 0.05.

**Figure 2 diagnostics-14-01945-f002:**
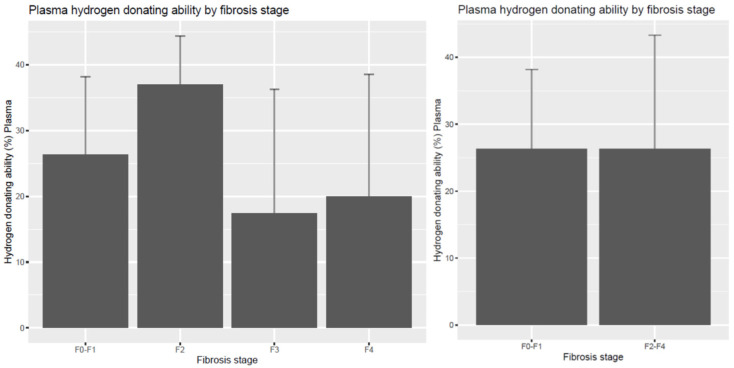
Hydrogen-donating ability of plasma (inhibition%) according to liver fibrosis stages. The compound plasma hydrogen-donating ability did not reflect the differences between the different fibrosis groups.

**Figure 3 diagnostics-14-01945-f003:**
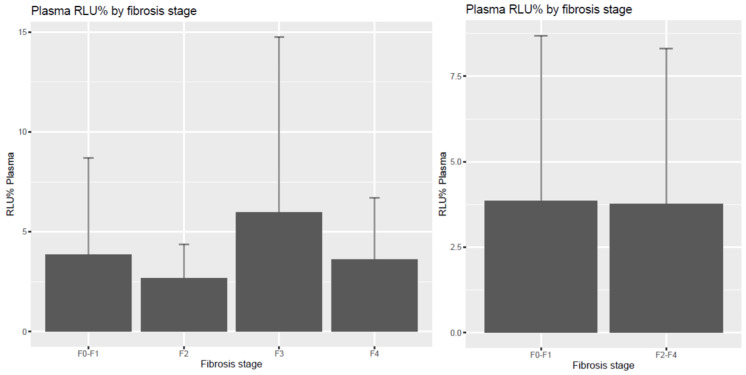
Induced free radical content in plasma (RLU%) according to liver fibrosis stages. Plasma total scavenger capacity, indicating the short-term pro-antioxidant balance, did not show significant changes in the advanced fibrosis groups.

**Figure 4 diagnostics-14-01945-f004:**
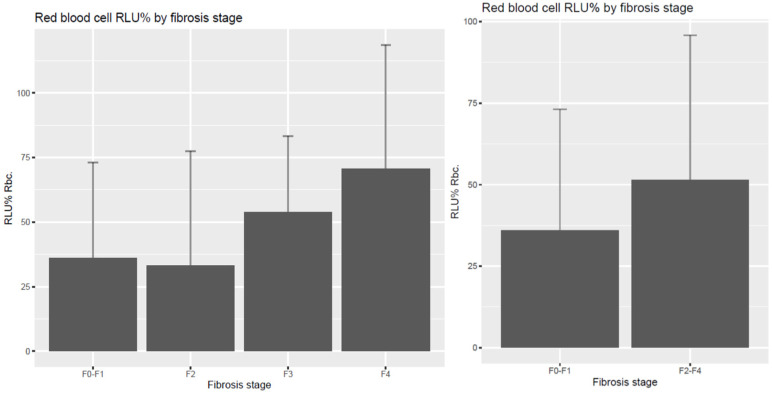
Induced free radical content in erythrocytes (RLU%) according to liver fibrosis stages. Increased erythrocyte chemiluminescence intensity, representing long-term impaired total scavenger capacity, was measured in the advanced fibrosis groups.

**Table 1 diagnostics-14-01945-t001:** Leptin, adiponectin, interleukin-6, and TNF-α concentrations (mean ± SD) according to liver fibrosis stages (F0–F4).

Dimension	F0–F1	F2	F3	F4	F2–4
Leptin (ng/mL)	4.88 ± 7.23	5.25 ± 5.60	4.51 ± 5.59	6.20 ± 7.23	5.45 ± 5.94
Adiponectin (pg/mL)	12,374.68 ± 13,658.2	9560.93 ± 6356.05	24,665 ± 21,830.25	16,810 ± 12,070.07	15,629.7 ± 13,801.38
Interleukin-6 (pg/mL)	1.36 ± 1.19	1.69 ± 1.65	2.21 ± 2.34	2.45 ± 2.93	2.12 ± 2.28
TNF-α (pg/mL)	78.91 ± 72.89	155.47 ± 151.01	41.07 ± 28.06	63.65 ± 57.59	94.34 ± 109.33

Higher cytokine concentrations were measured for the significant fibrosis groups without statistical significance.

## Data Availability

The data presented in this study are available on request from the corresponding author.

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
