# Peer review of "Redox Homeostasis and Non-Invasive Assessment of Significant Liver Fibrosis by Shear Wave Elastography"

_diagnostics, 2024, doi:10.3390/diagnostics14171945_

Round 1

Reviewer 1 Report

Comments and Suggestions for Authors

There are currently many manuscripts that take a similar approach to this one. This manuscript is also an interesting topic, but it contains some problems (see below). Thus, it is not acceptable for publication in the present form.

Major point

Please mention in the Discussion, citing previous papers, whether the diagnosis of fibrosis stage by shear wave elastography can be a gold standard.

Minor points

1.     In addition to each biomarker, are clinical findings not included in this study?

2.     Please indicate interoperator and intraoperator variability in the author's hospital for this shear wave elastography value.

3.     Please briefly describe the significance of each biomarker. Such explanations help the readers to understand this article more in depth.

Reviewer 2 Report

Comments and Suggestions for Authors

The article “The Redox Homeostasis and Non-Invasive Assessment of Significant Liver Fibrosis by Shear-Wave Elastography” by Egresi et al., is a cohort study (on 88 chronic liver disease patients) trying to examine and identify the best redox homeostasis biomarkers, and the liver stiffness measurements to better assess the significant liver fibrosis in chronic liver diseases of different etiologies.

The article has many imperfections that must be improved or corrected:

Line 31: Please, redefine the keywords according to the MeSH system.

Some keywords are already in the title and therefore should NOT have been listed as keywords in line 31.

Keywords: liver fibrosis; shear-wave elastography; cytokines; leptin; adiponectin; redox parameters

Line 41: historical standard – should be better explained!

Lines: 44-46: Please, reformulate:

Non-invasive methods also has a role in predicting complications and clinical out-

44

comes (esophageal varices formation or hepatocellular carcinoma development) and in

45

making a therapeutic and diagnostic decision.[1,2]

46

Line 46: .[1,2] - Please, rewrite all references correctly, moving the point outside the parentheses. NOT .[1,2]

Please, rewrite as: [1,2].

Lines: 51, 54: the same mistake!

Line 55: Please, reformulate the proposition with: "aspecificity" and explain better “METAVIR” in line 56.

Line 58: Please, rewrite as:  [9].

Line 59: Please, correct “play” instead of "plays" (plural).

Line 62: Please, reformulate: “liver....to be opened to free radicals”

Lines 65, 66 - Please, rewrite the points outside the brackets!

Lines 71-73: Please, reformulate the text “....with this value”. [13] 

Lines 73-74: .[1,14]

Lines 75-78: Please, explain better and correct the techniques and abbreviations, as well as put the references correctly.

Line 78: the final point is missing!

Line 82: TNF- …?

Line 84: USA

Line 95: kPa: Please, explain the abbreviation.

Lines 94-99 Please, reformulate better!

Line 118: a sentence is missing: "All..."? What does it mean here?

Lines 130-131: Please, reformulate correctly:

Data are expressen as

130

mean±SD.

131

Lines 133-136 - Please, reformulate the results!

1.      For the readers, Figures 1 - 4 need to be redesigned in a color-blind palette to better differentiate the information and the measurements.

2.      All figures should be better explained and discussed to meet the demands of a journal like Diagnostics.

Lines: 160-161 Please, correct the insertion of “Figure 3. Figure 4.” – see below:

Figure 3. Figure

160

4.

161

Line 166: 4. Discussion /s

Line 167: Please, reformulate: “Chronic liver diseases are significant worldwide problems”!?

Lines 167-168: Please, reformulate: “historical method”

Lines 169-171: Please reformulate: "However, it has many limitations, it is a painful, invasive procedure with many contraindications in a liver patient and the risk of complications, sampling errors and significant interobserver interpretation” and please, insert references!

Lines 171-172: "An increasing number of non-invasive markers have been investigated to predict the severity of liver fibrosis in the past decades". Please, insert references!

Lines 174, 175, 177 : Please, rewrite all references correctly.

Line 175-177 : Please reformulate:

The investi-
gation of non-invasive biomarkers are intensively focused by researchers, and new results

175
176

are regularly published. [1,14]

177

Line 182: Please, correct: Child-Pugh score. [25,26]

Idem line 185: . [12]

Line 190: The point is missing: for F4 patients [28] The Society of Radiologists…

Line 197: The final point is missing.

Line 212: …. in a 2024 meta-analysis. [32]

Idem, line 202: growth factor-β1. [35,36]

Idem, line 222: and fibrogenesis. [37]. The adiponec-

Line, 226: Please, correct:

well.[38]

226

Lines 228-229: Please, correct the punctuation and the references.

.

228

[39,40] We also

229

Lines 227-230: Please, reformulate the whole paragraph:

Some earlier studies showed a good correlation…and TNF- in the significant fibrosis groups
(F2-F4) without significance.”

Please, correct:

Line 239:

damage. [41]

239

Line 245:

“…in biological samples. [42] In our study…

Line 271:

as well.[42]

271

Line: 280  . [44]

Line 286:  . [45]

The conclusions should be improved.

The authors should imagine and draw a Graphical Abstract for this article that will increase the reader's understanding and visibility of the work.

A “List of Abbreviations” must be completed and reviewed carefully and may be better presented in a table format at the end. Any abbreviation must be entered in the first place where it was used and then reused as an abbreviation very judiciously.

A final reading on the MDPI platform by a native English speaker would be welcome.

Overall, I recommend a careful major revision!

Thank you very much!

July 10, 2024

Comments on the Quality of English Language

Moderate editing of English language required.

Reviewer 3 Report

Comments and Suggestions for Authors

Dear Authors, this is a good paper well written and elegant, exploring the significance of inflammatory pathways in liver diseases, according to fibrosis stages. Liver stiffness measured by pSWE is a good method but not the standard to evaluate the fibrosis. The paper should value as a "proof of concept" and i really understand the aim of the study beyond the results. However some limits there are. Please, modify NAFLD in steatotic diseases, and at line 228 there is an editorial issue (a space). 

Round 2

Reviewer 1 Report

Comments and Suggestions for Authors

This article is interesting, but the revision is insufficient.

Major points

1.     Please describe regarding body mass index, diabetes and alcohol intake. Do you exclude cases of hepatitis, cholecystitis, cholangitis, and liver tumors? What about cases with a history of previous malignancy treatment?

2.     Please indicate how you divided F0-1, F2, F3, and F4 and provide references to support this division.

3.     F2 (significant fibrosis) and F3,4 (severe fibrosis) should be considered separately.

4.     Is it possible to truly stratify F0-1 and F2? Please discuss deeply this problem in discussion.

 Minor points

5.  Please abbreviate the journal’s name according to the Author’s guideline.

6.  The following numbered journal names are missing, 19, 26, 27, 33, 34, 35, 40, 46. 

Comments on the Quality of English Language

 Please abbreviate the journal’s name according to the Author’s guideline.

 The following numbered journal names are missing, 19, 26, 27, 33, 34, 35, 40, 46.

Author Response

  1. Please describe regarding body mass index, diabetes and alcohol intake. Do you exclude cases of hepatitis, cholecystitis, cholangitis, and liver tumors? What about cases with a history of previous malignancy treatment?

BMI, alcohol consumption, occurance of hypertension and diabetes was added to the description (line 99, 107-111).

Many chronic liver disease patients with different origin were included-so hepatitis patients with viral reason (HBV, HCV, NASH, autoimmune), however acute cholecystitis, cholangitis, liver tumor patients, patients with previous malignancy history and therapy were excluded. We added a sentence regarding the inclusion and exclusion criterias (line 109-111).

  1. Please indicate how you divided F0-1, F2, F3, and F4 and provide references to support this division.

The cut-off values and the new reference was provided (line 122-124).

  1. F2 (significant fibrosis) and F3,4 (severe fibrosis) should be considered separately.

The results were focusing on F0-1 vs. F2-4 which is of most clinical significance (no and nonsignificant fibrosis vs significant fibrosis that demands intervention). We do not perform statistical analysis between the F2 vs. F3-4 fibrosis groups, because the the absence of clinical relevance. We added the abendment into the result part (line 165-166).

  1.      Is it possible to truly stratify F0-1 and F2? Please discuss deeply this problem in discussion.

Completion of the discussion was performed regarding the accuracy of elastography to stratify F0-F1 vs. F2 (line 256-261).

The missing journal names were complemented and the abbreviations were corrected.

Reviewer 2 Report

Comments and Suggestions for Authors

The authors tried to improve all the indicated aspects and I believe that the manuscript has improved.

1. As a first observation, the title of the article is still NOT written properly, in "Title Case" format.

2. The insertion of Figures 3 and 4 in the text is NOT appropriate, please, lines 194-195 should be corrected:

Please, see: .Figures 3. and 4. .............195

3. I believe that there are still many imperfections in the expression and the proofreading of the entire manuscript by a native English speaker on the MDPI platform, namely the Diagnostics journal, is absolutely necessary.

In conclusion, it is up to the Academic Editor to make the final decision to publish this article.

Thank you very much!

06 August 2024

Comments on the Quality of English Language

Proofreading of the entire manuscript by a native English speaker on the MDPI platform, namely the Diagnostics journal, is absolutely necessary.

Author Response

  1. As a first observation, the title of the article is still NOT written properly, in "Title Case" format.

The title is rewritten in “Title case” format

  1. The insertion of Figures 3 and 4 in the text is NOT appropriate, please, lines 194-195 should be corrected:

Please, see: .Figures 3. and 4. .............195

The insertion was corrected.

Reviewer 3 Report

Comments and Suggestions for Authors

Dear Authors the paper, already evaluated by me, is a good proof of concept as stated. Thank You for your replies, that are coherent with our requests. Anyway, the absence of the biopsies is a real lack to confirm Your results. 

Author Response

Dear Authors the paper, already evaluated by me, is a good proof of concept as stated. Thank You for your replies, that are coherent with our requests. Anyway, the absence of the biopsies is a real lack to confirm Your results. 

We emphasized the lack of liver biopsy as a limitation of the study (line 359-365).

Round 3

Reviewer 1 Report

Comments and Suggestions for Authors

This manuscript has been fully revised and acceptable for publication in present form.

Reviewer 3 Report

Comments and Suggestions for Authors

You replied correctly to our answers. Particularly, You underlined the question about the lack of liver biopsy